# Global Trends (1961–2017) in Human Dietary Potassium Supplies

**DOI:** 10.3390/nu13041369

**Published:** 2021-04-19

**Authors:** Diriba B. Kumssa, Edward J. M. Joy, Martin R. Broadley

**Affiliations:** 1Sutton Bonington Campus, School of Biosciences, University of Nottingham, Nottinghamshire LE12 5RD, UK; martin.broadley@nottingham.ac.uk; 2Faculty of Epidemiology and Population Health, London School of Hygiene & Tropical Medicine, Keppel Street, London WC1E 7HT, UK; edward.joy@lshtm.ac.uk

**Keywords:** adequate intake, potassium sufficiency ratio, potassium to sodium ratio, recommended daily allowance

## Abstract

Background: Potassium (K) is an essential mineral and major intracellular electrolyte involved in the regulation of blood pressure, muscle contraction and nerve transmission in humans. Major dietary sources of K include fruits and vegetables, starchy roots and tubers, and whole grains. The aim of this study was to assess and report: (i) the sufficiency of K in national food systems globally, (ii) to quantify the contribution from food groups, and (iii) to explore spatial and temporal trends in the period of 1961–2017. Methods: Food supply and demography (1961–2017), K composition and K requirement data were combined to estimate per capita human dietary supplies of potassium (DSK), adequate intake of K (AIK) and K sufficiency ratio (KSR) at national, regional, continental and global levels. Results and Discussion: Globally, the mean ± SD. DSK (mg capita^−1^ d^−1^) increased from 2984 ± 915 in 1961 to 3796 ± 1161 in 2017. There was a wide range in DSK between geographical regions and across years, with particularly large increases in east Asia, where DSK increased from <3000 to >5000 mg capita^−1^ day^−1^. Roots and tubers contributed the largest dietary source of K, providing up to 80% of DSK in most regions. At the global level, throughout the 57-year period, the population-weighted KSR was <1 based on the 2006 Institute of Medicine AIK recommendation, while it was >1 based on the 2019 National Academies of Science and the 2016 European Union AIK recommendation. While KSR ≥ 1 shows sufficiency of DSK, KSR < 1 does not indicate K deficiency risk. Conclusion: Due to the absence of a Recommended Daily Allowance (RDA) for K, this study used the ratio of DSK:AIK (i.e., KSR) to assess dietary K sufficiency. Estimates of dietary K sufficiency are, therefore, highly sensitive to the AIK reference value used and this varied greatly based on different institutions and years. To quantify the risk of dietary K deficiency, bridging the data gap to establish an RDA for K should be a global research priority.

## 1. Introduction

Potassium (K) is an essential mineral for human health [1,2]. It plays vital roles in the normal functioning of cells and organs [3,4] through its involvement in nerve transmission, muscle contractions, regulation of blood pressure and maintenance of the integrity of the skeleton [1,5,6,7,8]. The major dietary sources of K are fruits, legumes, starchy roots and tubers, whole grains, and vegetables [2,9,10,11]. Homeostatic control of serum K concentration occurs mainly through the regulation of urinary K excretion. However, reduced dietary intake of Mg [12] and K [2], and increased consumption of Na [13,14] due to food processing and high intake of salt [15] affect the utilisation of K. Loss of K during food processing reduces dietary K intake [16,17,18].

Several studies have assessed dietary K intakes and risks of deficiency through cross-sectional dietary surveys. For example, according to the Dietary Guideline for Americans, in 2015, K was one of the dietary nutrients typically under-consumed and of public health concern in the United States [19]. A cross-sectional study in the UK in 2018 found a negative association between dietary K concentration and the proportion of processed foods in the diet, and 90% of the population did not meet the recommended K intake [14].

Dietary requirement reference values for K to assess K deficiency risks in populations have not been established due to insufficient evidence [9]. However, Adequate Intakes (AI) have been proposed by the Institute of Medicine (IOM) in 2006 [9] and revised in 2019 by the National Academies of Sciences, Engineering, and Medicine [20], in which values were defined as the median K intake observed in apparently healthy individuals in the US and Canada. An AI is “the recommended daily intake of a nutrient estimated by the Institute of Medicine to meet or exceed the amount needed to maintain adequate nutrition for most people in a particular life stage and gender group”. An AI may be set for nutrients where there is insufficient evidence to determine a Recommended Dietary Allowance (RDA), which is the average daily intake level sufficient to meet the requirements of nearly all (97.5%) healthy individuals in an age- and sex-specific population group [9]. Turck et al. [2] have proposed AI values for K for Europe, while the World Health Organization (WHO) [10] has separate global recommendations (see Table 1). Adequate Intake values for K are derived from dietary intake levels that maintain normotensive blood pressure (BP) and reduce the impact of salt on BP [2,10,20].

The prevalence of K deficiency in populations with median K supply greater or equal to the AI is likely to be low; however, the inverse cannot be assumed where the median K intake is less than the AI [21], and this limits the ability to assess the adequacy of population K intakes. This contrasts with other mineral micronutrients which have RDAs and Estimated Average Requirement (EAR) values defined, from which the prevalence of inadequate dietary intakes and risk of deficiency can be estimated [9,12,21,22,23].

The aim of this study was to assess and report the sufficiency of K in national food systems globally, to quantify the contribution from food groups, and to explore spatial and temporal trends over recent decades. The study compares per capita dietary K supplies in national food systems to population-weighted AIs and discusses likely implications for risks of deficiency.

## 2. Materials and Methods

Per capita human dietary supplies of potassium (DSK), adequate intake of K (AIK) and the ratio between DSK and AIK, i.e., K sufficiency ratio (KSR), were estimated at national, regional, and global levels. Dietary K contributions from various food groups (i.e., animal products, cereals, fruits and vegetables, pulses and beans, roots and tubers and others) to DSK were estimated at the regional level between 1961 and 2017. Demographic data were used to estimate population-weighted DSK, AIK, and KSR at national, regional, continental and global levels.

### 2.1. Demographic Data

Demographic data by age and sex (1961–2017) were downloaded from the United Nations World Population Prospects website [24]. Data on births by age of mother were used to estimate the number of pregnant and lactating women during a given year. The number of pregnant and lactating women each year was estimated using Equations (1) and (2). Pregnancy was assumed to last for nine months and mothers were assumed to breastfeed for 1.5 years. Demographic data were used to estimate the population-weighted DSK, AIK and KSR at various geographical scales (see Appendix A). Prior to weighting DSK, AIK and KSR by population, the number of pregnant and lactating women of the five-year age groups in the reproductive ages (i.e., 15–49 years) was subtracted from the total number of women in the corresponding age groups.
(1)Number of pregnant women=number of births×0.75
where number of births was assumed to be equivalent to the number of women in a five-year reproductive age group, and pregnancy lasted for nine months.
(2)Number of lactating women=number of births×1.5
where breastfeeding (lactation) was assumed to last for one and half years on average.

### 2.2. Dietary Supplies of Potassium (DSK)

Daily per capita plant and animal source food supply data for 57 years (1961–2017) were downloaded from the Food and Agriculture Organization (FAO) of the United Nations statistical data (FAOSTAT) [25] website. The daily per capita food supply statistic was considered as a proxy for daily food intake. Potassium concentration data for each food item were obtained from the United States Department of Agriculture (USDA) Nutrient Data Laboratory (NDL) (Standard Reference 26) [26]. Per capita dietary supplies of K (DSK) were estimated by multiplying the daily food supply by the concentration of K in each food item. Demographic data were used to derive a population-weighted dietary supplies of K (WtdDSK) at regional, continental and global levels [22]. Dietary K contributions from various food groups were also estimated at regional levels between 1961 and 2017.

### 2.3. Potassium Requirement

Various institutions have set Adequate Intake (AI) values, and these are updated as new evidence emerges (see Table 1). In this study, AI values of K (AIK) for age- and sex-specific groups were derived from the Institute of Medicine (IOM) 2006 K requirement recommendation (AIK_06) [9], National Academies of Sciences (NAS) 2019 K requirement recommendation (AIK_19) [20], and the K requirement recommendation for the European Union (AIK_EU) [2]. A population-weighted AI of K was calculated (WtdAIK) at national, regional, continental, and global levels using annual demographic information. The impact of source of AIK recommendation on the risk of dietary K deficiency was compared.

### 2.4. Potassium (K) Sufficiency Ratio

The Potassium (K) Sufficiency Ratio (KSR) between dietary supplies of K (DSK) and the population-weighted adequate intake of K (WtdAIK) was calculated as indicated in Equation (3). A KSR ≥ 1 indicates that there is likely to be low risk of K deficiency due to inadequate dietary supplies in populations at aggregated scales, i.e., national, regional, continental, and global levels. Demographic data were used to derive a population-weighted dietary supplies of potassium (WtdDSK) and potassium adequacy ratio (WtdKSR) at regional, continental and global levels.
(3)$$KSR=\frac{DSK}{AIK}$$
where *KSR* is potassium sufficiency ratio, *DSK* is dietary supplies of *K*, and *AIK* is recommended adequate intake of *K*.

### 2.5. Data Analyses and Visualisation

Data compilation and management were carried out using Microsoft Office 365 Access 2016. Regional-, continental-, and global-level aggregations, weighting by population and derivation of descriptive statistics were conducted using IBM^©^ SPSS Statistics version 27 [27]. Thematic maps were produced using QGIS 3.16 Hannover [28]. Line graphs were produced using the ggplot2 package [29] in R version 4.0.2 [30].

## 3. Results

### 3.1. Dietary Potassium (K) Supplies and Requirements

At the national level, in 1961, dietary supplies of K (DSK) ranged from 1062 mg capita^−1^ day^−1^ in Myanmar to 7016 mg capita^−1^ day^−1^ in Côte d’Ivoire. In 1997, DSK ranged from 1285 mg capita^−1^ day^−1^ in Cambodia to 6428 mg capita^−1^ day^−1^ in Ghana. In 2017, DSK ranged from 1705 mg capita^−1^ day^−1^ in The Gambia to 7372 mg capita^−1^ day^−1^ in Ghana (Appendix A and Figure 1). In 1961, population-weighted adequate intake (AI) of K (WtdAIK) ranged from 4204 mg capita^−1^ day^−1^, 2553 mg capita^−1^ day^−1^ and 2661 mg capita^−1^ day^−1^ in Grenada to 4509 mg capita^−1^ day^−1^, 2802 mg capita^−1^ day^−1^ and 3157 mg capita^−1^ day^−1^ in Sweden based on AIK_06, AIK_19 and AIK_EU values, respectively. In 1997, the WtdAIK ranged from 4213 mg capita^−1^ day^−1^, 2574 mg capita^−1^ day^−1^ and 2665 mg capita^−1^ day^−1^ in Yemen to 4572 mg capita^−1^ day^−1^, 2855 mg capita^−1^ day^−1^ and 3271 mg capita^−1^ day^−1^ in Italy based on AIK_06, AIK_19 and AIK_EU values, respectively. In 2017, the WtdAIK ranged from 4209 mg capita^−1^ day^−1^, 2571 mg capita^−1^ day^−1^, and 2663 mg capita^−1^ day^−1^ in Niger to 4587 mg capita^−1^ day^−1^, 2876 mg capita^−1^ day^−1^ and 3299 mg capita^−1^ day^−1^ in Japan based on AIK_06, AIK_19 and AIK_EU values, respectively (Appendix A).

At the regional level, in 1961, the population-weighted mean dietary supplies of K (WtdDSK) ± standard deviation (SD) ranged from 1798 ± 336 mg capita^−1^ day^−1^ in south-eastern Asia to 4837 ± 936 mg capita^−1^ day^−1^ in eastern Europe. In 1997, the mean WtdDSK ± SD ranged from 2067 ± 284 mg capita^−1^ day^−1^ in south-eastern Asia to 4744 ± 350 mg capita^−1^ day^−1^ in southern Europe. In 2017, the mean WtdDSK ± SD ranged from 2744 ± 434 mg capita^−1^ day^−1^ in south-eastern Asia to 5262 ± 799 mg capita^−1^ day^−1^ in eastern Asia (Figure 2 and Appendix A). At the regional level, in 1961, the population-weighted mean adequate intake of K (WtdAIK) ± SD ranged from 4216 mg capita^−1^ day^−1^, 2577 mg capita^−1^ day^−1^ and 2617 mg capita^−1^ day^−1^ in Polynesia to 4486 ± 19 mg capita^−1^ day^−1^, 2779 ± 17 mg capita^−1^ day^−1^ and 3121 ± 35 mg capita^−1^ day^−1^ in northern Europe based on AIK_06, AIK_19 and AIK_EU values, respectively. In 1997, the mean WtdAIK ± SD ranged from 4254 ± 23 mg capita^−1^ day^−1^, 2601 ± 14 mg capita^−1^ day^−1^ and 2733 ± 36 mg capita^−1^ day^−1^ in central Africa to 4561 ± 26 mg capita^−1^ day^−1^, 2843 ± 23 mg capita^−1^ day^−1^ and 3250 ± 46 mg capita^−1^ day^−1^ in southern Europe based on AIK_06, AIK_19 and AIK_EU values, respectively. In 2017, the mean WtdAIK ± SD ranged from 4270 ± 26 mg capita^−1^ day^−1^, 2612 ± 22 mg capita^−1^ day^−1^ and 2758 ± 43 mg capita^−1^ day^−1^ in central Africa to 4577 ± 9 mg capita^−1^ day^−1^, 2864 ± 7 mg capita^−1^ day^−1^ and 3278 ± 15 mg capita^−1^ day^−1^ based on AIK_06, AIK_19 and AIK_EU values, respectively (Figure 2 and Appendix A).

At the continental level, in 1961, the population-weighted mean (WtdDSK) ± SD dietary supplies of K ranged from 2498 ± 536 mg capita^−1^ day^−1^ in Asia to 4458 ± 569 mg capita^−1^ day^−1^ in Europe. In 1997, the mean WtdDSK ± SD ranged from 3121 ± 852 mg capita^−1^ day^−1^ in Asia to 4368 ± 432 mg capita^−1^ day^−1^ in Europe. In 2017, the mean WtdDSK ± SD ranged from 3417 ± 330 mg capita^−1^ day^−1^ in Oceania to 4064 ± 1160 mg capita^−1^ day^−1^ in Africa (Appendix A). At the continental level, in 1961, the population weighted mean adequate intake of K (WtdAIK) ± SD ranged from 4279 ± 26 mg capita^−1^ day^−1^, 2628 ± 20 mg capita^−1^ day^−1^ and 2782 ± 41 mg capita^−1^ day^−1^ in Africa to 4466 ± 34 mg capita^−1^ day^−1^, 2762 ± 25 mg capita^−1^ day^−1^ and 3089 ± 58 mg capita^−1^ day^−1^ in Europe based on AIK_06, AIK_19 and AIK_EU values, respectively. In 1997, the mean WtdAIK ± SD ranged from 4295 ± 52 mg capita^−1^ day^−1^, 2627 ± 36 mg capita^−1^ day^−1^ and 2797 ± 80 mg capita^−1^ day^−1^ in Africa to 4545 ± 19 mg capita^−1^ day^−1^, 2821 ± 22 mg capita^−1^ day^−1^ and 3217 ± 34 mg capita^−1^ day^−1^ in Europe based on AIK_06, AIK_19 and AIK_EU values, respectively. In 2017, the mean WtdAIK ± SD ranged from 4316 ± 58 mg capita^−1^ day^−1^, 2649 ± 50 mg capita^−1^ day^−1^ and 2836 ± 98 mg capita^−1^ day^−1^ in Africa to 4556 ± 19 mg capita^−1^ day^−1^, 2845 ± 20 mg capita^−1^ day^−1^ and 3246 ± 32 mg capita^−1^ day^−1^ in Europe based on AIK_06, AIK_19 and AIK_EU values, respectively (Appendix A).

At the global level, the population-weighted mean dietary supplies of K (WtdDSK) ± SD ranged from 2984 ± 915 mg capita^−1^ day^−1^ in 1961 to 3796 ± 1161 mg capita^−1^ day^−1^ in 2017 (Appendix A). Global WtdDSK values were well above the population-weighted AI of K (WtdAIK) based on recommendations of the AIK_19 and AIK_EU but below recommendations of AIK_06 (Figure 3).

Across the 57-year period, roots and tubers contributed the largest dietary source of K, providing up to 80% in all regions except northern Africa, central America, south-eastern Asia, southern Asia and western Asia (Figure 4). Animal source foods contributed much smaller proportions of dietary K supplies than cereals, pulses and beans, and fruits and vegetables in all regions (Figure 4).

### 3.2. Potassium Sufficiency Ratio (KSR)

At the national level, in 1961, out of 138 countries, 118, 49 and 54 countries had K sufficiency ratio (KSR) < 1 based on the AIK_06, AIK_19 and AIK_EU K requirement recommendation, respectively. In 1997, out of 160 countries, 129, 36 and 47 countries had KSR < 1 based on the AIK_06, AIK_19 and AIK_EU K requirement recommendation, respectively. In 2017, out of 164 countries, 139, 36 and 56 countries had KSR < 1 based on the AIK_06, AIK_19 and AIK_EU K requirement recommendation, respectively (Figure 1 and Appendix A).

At the regional level, in 1961, out of the 21 regions, 17, 3 and 4 regions had population-weighted K sufficiency ratio (WtdKSR) < 1 based on the AIK_06, AIK_19 and AIK_EU K requirement recommendation, respectively. In 1997, out of the 22 regions, 21, 2 and 3 regions had WtdKSR < 1 based on the AIK_06, AIK_19 and AIK_EU K requirement recommendation, respectively. In 2017, out of the 22 regions, 19, 1 and 3 regions had WtdKSR < 1 based on the AIK_06, AIK_19 and AIK_EU K requirement recommendation, respectively (Figure 5 and Appendix A).

At the global level, throughout the 57-year period, the WtdKSR was <1 based on the AIK_06 K requirement recommendation, while it was >1 based on the AIK_19 and AIK_EU K requirement recommendation (Figure 4 and Appendix A).

## 4. Discussion

There was a wide range in dietary supplies of K between countries, regions and continents. Some regions saw an increase in dietary K supplies in the period 1961–2017, most notably East Asia, where the regional supplies (a population-weighted mean of national supplies) increased from <3000 to >5000 mg capita^−1^ day^−1^. Other regions have seen less dramatic changes, and dietary K supplies have fallen over the period of 1961–2017 in European Regions by approximately 7–21%.

Dietary K supplies appear to be sufficient at a national level for most countries, when compared against population-weighted AIs. This is indicative of adequate K supplies in national food systems to meet population requirements. However, this comparison is very sensitive to the source of dietary reference values. This is illustrated by comparing dietary K supplies against AI values set by the IOM in 2006 [9], which were ~30–40% greater for most demographic groups than the values proposed more recently by the National Academies of Science (NAS) [20] and the European Union (EU) [2]. With the availability of more evidence, the AI values may be updated again. In the future, it may be possible to propose the Estimated Average Requirement (EAR) and Recommended Daily Allowance (RDA) reference values, which will enable the estimation of the prevalence of inadequate dietary K supplies at national levels using an adapted EAR cut-point methodology, as described previously for other micronutrients [22,23,31,32].

Dietary reference values are defined for healthy populations, and there may be groups of individuals at risk of K deficiency despite adequate K supplies in national food systems. Use of diuretics, for example, in the treatment of high blood pressure and heart disease can lead to K deficiency, and K supplementation may be required [33,34]. Taking K supplements may also reduce blood pressure for individuals with hypertension [35], although the evidence is not conclusive [36], and excessive supplementation with K may increase blood pressure in some individuals [37]. In addition, nutrient imbalances, especially high intake of sodium (Na), negatively affect the utilisation of K by the body. The WHO recommends a dietary K:Na ratio of 1:1 [38]; however, prevailing K:Na ratios in diets are substantially lower (i.e., K intake less than Na intake) [3,39,40]. In agreement with previous studies, for example, in the current study, at an estimated global average Na intake [41] of 4 g capita^−1^ day^−1^, the global K:Na ratio in 1961 and 2017 was 373:500 and 949:1000, respectively. Similarly, based on the National Health and Nutrition Examination Survey (NHANES) 2017–2018 nutrient intake in the USA [42], the K:Na intake ratio for adult female and male population aged 20–29 years was 2643:4219 and 2309:3326, respectively.

This study estimated dietary K supplies using food supply data from the FAOSTAT which has inherent limitations. Food Balance Sheets (FBS) may over-estimate the availability of foods and nutrients since they do not account for food loss or waste at the household level, or they may underestimate certain food items by failing to capture subsistence production [43]. In addition, the aggregation of some food items (e.g., cereals, vegetables, fruits, etc.) in the FAOSTAT makes it difficult to match K concentration in some food items in the United States Department of Agriculture Nutrient Data Laboratory Dietary food list. Similarly, K concentration in food varies across years, locations and cultivars of a given species which cannot be accounted for due to unavailability of food composition data with better temporal and spatial resolution. These can be sources of uncertainty in our analyses and conclusions, and our results should be interpreted with due caution. As an alternative to FBS, dietary K supplies may be estimated with dietary surveys using recall or food diary methods which are available for nationally representative populations or large cohorts in some countries. For example, estimated K intakes among adult males and females aged 20–29 years in the 2017–2018 NHANES survey were 2643 and 2309 mg capita^−1^ day^−1^, respectively [42]. This compares to a national dietary supply estimate from FBS of 3534 mg capita^−1^ day^−1^ for the United States in 2017. It is common to find greater dietary supply estimates when using FBS compared to individually administered dietary surveys, and this is likely due to limitations in the FBS data as well as limitations in dietary surveys in which respondents often under-report food consumption [44,45].

In the present study, food composition data were derived from the USDA which provides a high-quality source of data. However, the composition of crops including staple cereals shows spatial variability for some mineral micronutrients [46,47]. Whether this is the case for K, and the extent to which this affects estimates of dietary supply and adequacy will be important to establish but is considered outside the scope of the current study. Furthermore, a concerted research effort to close the data gap to establish the RDA for K is very crucial to enable accurate assessment of the risk of dietary K deficiency.

## 5. Conclusions

Due to the absence of Recommended Daily Allowance (RDA) for K, this study used the ratio of DSK:AIK, i.e., KSR to assess dietary K sufficiency. While KSR > 1 shows sufficiency of DSK, KSR < 1 does not indicate K deficiency risk. Furthermore, the sufficiency of DSK is contingent upon the AIK reference value used which varied greatly across institutions and years. To quantify the risk of dietary K deficiency, bridging the data gap to establish an RDA for K should be a global research priority.

## Figures and Tables

**Figure 1 nutrients-13-01369-f001:**
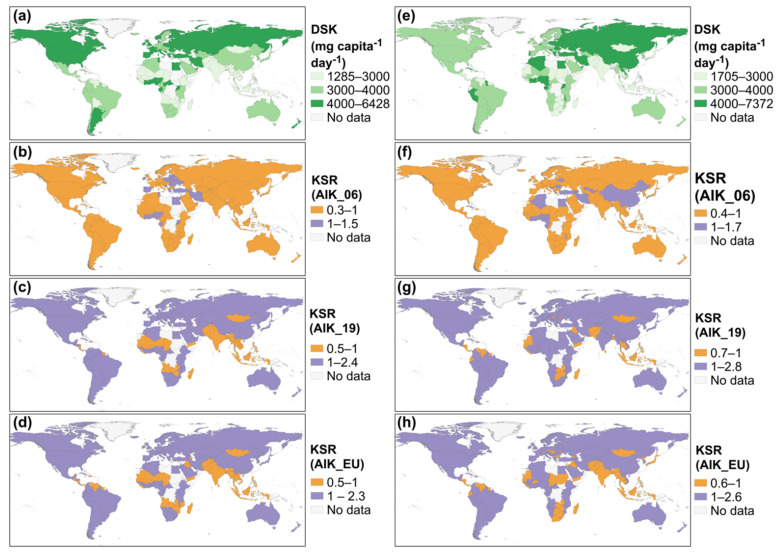
National-level dietary supplies of K (DSK), and K sufficiency ratio (KSR) in (**a**–**d**) 1997 and (**e**–**h**) 2017. KSR was calculated using the K AI recommendations by: (**b**,**f**) the Institute of Medicine in 2006 (AIK_06), (**c**,**g**) National Academies of Sciences in 2019 (AIK_19), and (**d**,**h**) the European Union (AIK_EU).

**Figure 2 nutrients-13-01369-f002:**
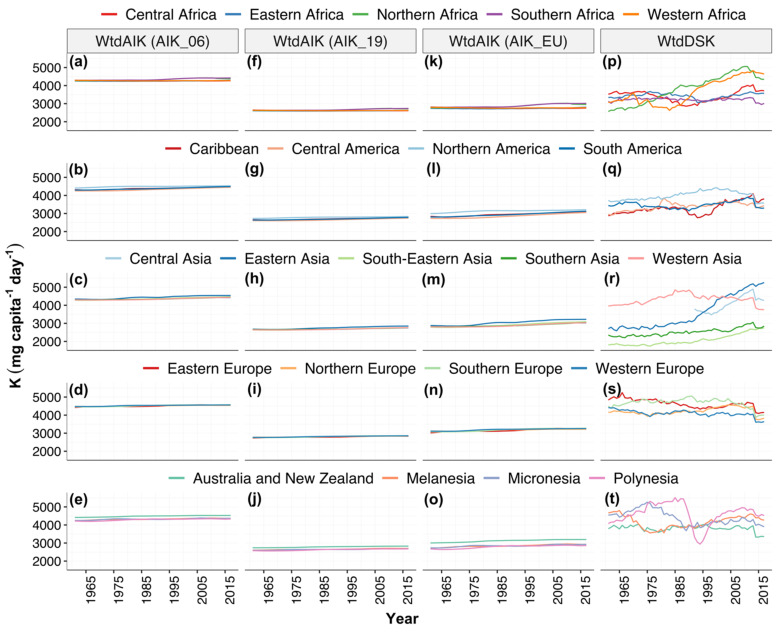
Regional-level population-weighted adequate intake of K (WtdAIK) and dietary supplies of K (WtdDSK), 1961–2017. Population-weighted adequate intake of K (WtdAIK) calculated using (**a**–**e**) the AI values of the Institute of Medicine in 2006 (**AIK_06**), (**f**–**j**) National Academies of Sciences in 2019 (AIK_19), and (**k**–**o**) the European Union (AIK_EU). (**p**–**t**) Population-weighted dietary potassium supplies of K (WtdDSK).

**Figure 3 nutrients-13-01369-f003:**
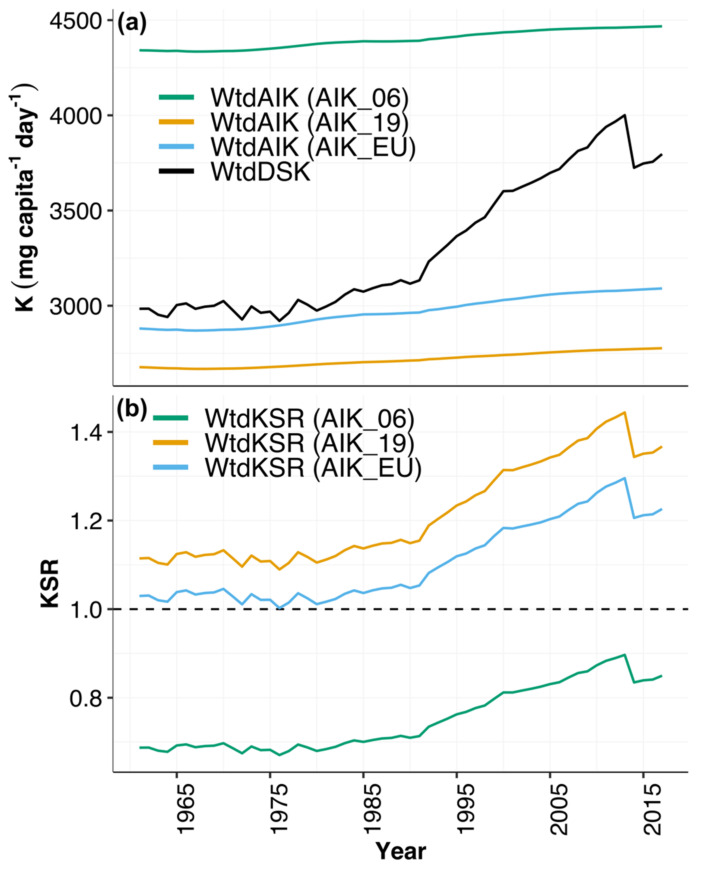
Global-level population-weighted (**a**) K supplies (WtdDSK), adequate intake (WtdAIK) and (**b**) K sufficiency ratio (WtdKSR) 1961–2017. WtdAIK was calculated using the AI values reported by the Institute of Medicine in 2006 (AIK_06), National Academies of Sciences in 2019 (AIK_19) and the European Union (AIK_EU). KSR was calculated using the three sources of AIK. The dashed black line represents KSR = 1.

**Figure 4 nutrients-13-01369-f004:**
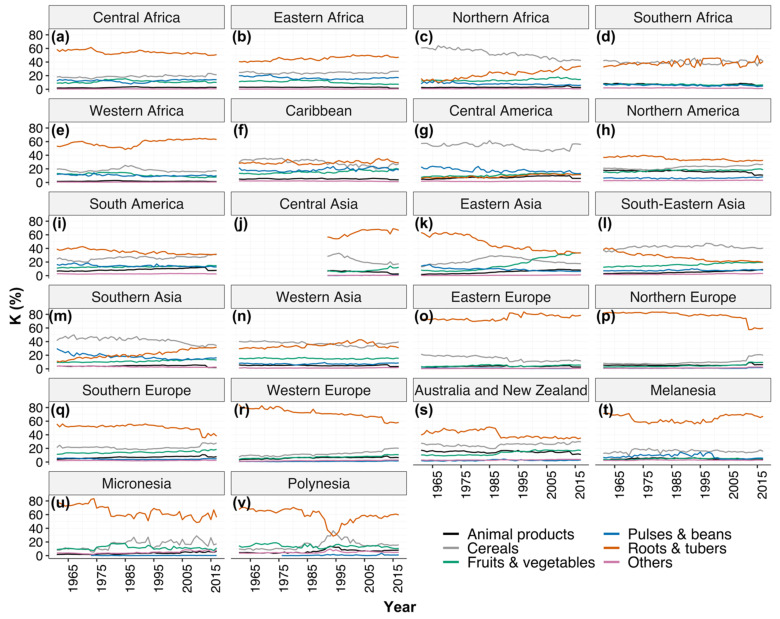
Percentage contribution of various food groups to dietary supplies of K in the different regions in: (**a**–**e**) Africa, (**f**–**i**) Americas, (**j**–**n**) Asia, (**o**–**r**) Europe and (**s**–**v**) Oceania.

**Figure 5 nutrients-13-01369-f005:**
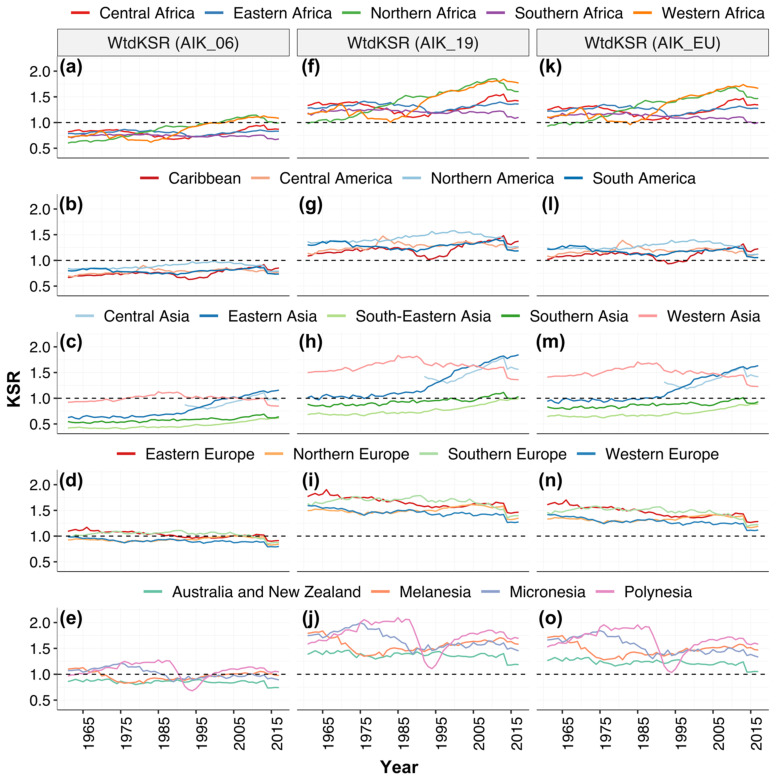
Regional-level population-weighted K sufficiency ratio (WtdKSR), 1961–2017. KSR was calculated using the recommended AI by: (**a**–**e**) the Institute of Medicine in 2006 (AIK_06), (**f**–**j**) National Academies of Sciences in 2019 (AIK_19), and (**k**–**o**) the European Union (AIK_EU).

**Table 1 nutrients-13-01369-t001:** Recommended adequate intake of potassium (AIK) by the Institute of Medicine (IOM) in 2006 (AIK_06) [9], and National Academies of Sciences (NAS) in 2019 (AIK_19) [20]; the European Union (AIK_EU) [2], and a conditional recommendation by the World Health Organization (WHO) [10].

Life-Stage Group	Age	AIK (mg capita^−1^ day^−1^)			
		AIK_06	AIK_19	AIK_EU	WHO
Infants	0–6 months	400	400		
	7–12 months	700	860	750	
Children	1–3 years	3000	2000	1100	
	4–8 years	3800	2300	1800	
Males	9–13 years	4500	2500	2700	
	14–18 years	4700	3000	3500	3510
	19–30 years	4700	3400	3500	3510
	31–50 years	4700	3400	3500	3510
	51–70 years	4700	3400	3500	3510
	>70 years	4700	3400	3500	3510
Females	9–13 years	4500	2300	2700	
	14–18 years	4700	2300	3500	3510
	19–30 years	4700	2600	3500	3510
	31–50 years	4700	2600	3500	3510
	51–70 years	4700	2600	3500	3510
	>70 years	4700	2600	3500	3510
Pregnancy	14–18 years	4700	2600	3500	
	19–30 years	4700	2900	3500	
	31–50 years	4700	2900	3500	
Lactation	14–18 years	5100	2500	4000	
	19–30 years	5100	2800	4000	
	31–50 years	5100	2800	4000	

## Data Availability

Data are provided as Appendix A.

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
