# Peer review of "Global Trends (1961–2017) in Human Dietary Potassium Supplies"

_nutrients, 2021, doi:10.3390/nu13041369_

Round 1
Reviewer 1 Report
A very interesting work of great importance in terms of public health. The methodology of the work is not in doubt and may be of use to other researchers designing their research on consumption and temporal trends. The work provides interesting and very important knowledge about global trends in potassium consumption and its changes. This is especially important when it comes to nutrient deficiencies and making recommendations to the public. Methodology and discussion are the strengths of the work.
Author Response
We would like to thank the reviewer for the compliments about the manuscript.
Reviewer 2 Report
The study by Kumssa et al., seeks to assess the sufficiency of potassium in food systems globally and to quantify the contribution of various foods to potassium intake over the period of 1961-2017. A lot of data is presented but it is laborious to get through and at times, seems like a regurgitation of numbers. The figures are not easy to look at and it is difficult to interpret the different colors on the graphs as well as the color associated with the key at the top. Finding a way to better present the data in the first few paragraphs of the results would be helpful as well as updating the figures. Overall, there is a lot of data presented and the readability of the results is low. Some grammatical improvements are needed as well. Overall, I think it is important information but hard to get through.
Title I would remove the abbreviation for potassium from the title. Not necessary.
Abstract
I would remove the numbers in the front of the subheadings of methods, results, etc. You already have the subtitles to separate the section
Not necessary to include “Globally, the mean +/- SD”. The readers will see this in your methods section.
Line 18-19 include units for potassium values
Line 22 what does this “WtdKSR” stand for?
Intro
Line 35 remove the word “element”. Mineral is sufficient
Line 41 and 42. Sodium and salt is separated however one can argue that most sodium in the diet is in the form of salt.
Methods
The IOM recommendations for potassium were updated in 2019. I would acknowledge that and state why they were not used
Results
Overall, much of the data in the text appears to just be regurgitation of numbers with reference to supplemental tables. I’m not sure what the main point is with relisting this information.
The figures looks unprofessional. Consider finding a better graphing program to use. It is difficult to see the colors of the different lines in the graphs as well as the key. In some figures, the key up top is so small, it is unclear what color belongs to which country or region
Author Response
Point by point tabulated response is attached.
